# Current Insights into the Effects of Dietary α-Linolenic Acid Focusing on Alterations of Polyunsaturated Fatty Acid Profiles in Metabolic Syndrome

**DOI:** 10.3390/ijms25094909

**Published:** 2024-04-30

**Authors:** Marija Takić, Slavica Ranković, Zdenka Girek, Suzana Pavlović, Petar Jovanović, Vesna Jovanović, Ivana Šarac

**Affiliations:** 1Centre of Research Excellence in Nutrition and Metabolism, Group for Nutrition and Metabolism, National Institute of Republic of Serbia, Institute for Medical Research, University of Belgrade, Tadeuša Košćuska 1, 11000 Belgrade, Serbia; vica0282@gmail.com (S.R.); suzapavlovic@gmail.com (S.P.); petar.jovanovic@imi.bg.ac.rs (P.J.); ivana.sarac@imi.bg.ac.rs (I.Š.); 2Department of Biochemistry and Centre of Excellence for Molecular Food Sciences, Faculty of Chemistry, University of Belgrade, Studentski trg 12-16, 11158 Belgrade, Serbia; vjovanovic@chem.bg.ac.rs

**Keywords:** ALA, n-3 PUFA, desaturase, oxylipins

## Abstract

The plant-derived α-linolenic acid (ALA) is an essential n-3 acid highly susceptible to oxidation, present in oils of flaxseeds, walnuts, canola, perilla, soy, and chia. After ingestion, it can be incorporated in to body lipid pools (particularly triglycerides and phospholipid membranes), and then endogenously metabolized through desaturation, elongation, and peroxisome oxidation to eicosapentaenoic acid (EPA) and docosahexaenoic acid (DHA), with a very limited efficiency (particularly for DHA), beta-oxidized as an energy source, or directly metabolized to C18-oxilipins. At this moment, data in the literature about the effects of ALA supplementation on metabolic syndrome (MetS) in humans are inconsistent, indicating no effects or some positive effects on all MetS components (abdominal obesity, dyslipidemia, impaired insulin sensitivity and glucoregulation, blood pressure, and liver steatosis). The major effects of ALA on MetS seem to be through its conversion to more potent EPA and DHA, the impact on the n-3/n-6 ratio, and the consecutive effects on the formation of oxylipins and endocannabinoids, inflammation, insulin sensitivity, and insulin secretion, as well as adipocyte and hepatocytes function. It is important to distinguish the direct effects of ALA from the effects of EPA and DHA metabolites. This review summarizes the most recent findings on this topic and discusses the possible mechanisms.

## 1. Introduction

α-linolenic acid (ALA, 18:3n-3) is an essential fatty acid found in plants that can be endogenously metabolized to its elongation/desaturation products eicosapentaenoic (EPA, 20:5n-3), docosapentaenoic n-3 (DPA n-3, 22:5n-3) and docosahexsaenoic acid (DHA, 22:6n-3), through a series of desaturation, elongation, and beta-oxidation steps [1,2]. ALA is essential because of the lack of delta-15 desaturase required for its synthesis [3]. Dietary-rich sources of this fatty acid (FA) include canola and soybean oils, flaxseed, walnuts, perilla, chia, *Camelina sativa* and other plant food sources [2,4,5]. ALA is the most important n-3 source in the diets of people who do not regularly consume oily fish (not only vegetarians and vegans) or take EPA and DHA supplements. The recommended daily intake for women is 1.1 g and for men 1.6 g to maintain adequate nutrition [6]. However, the potentially adverse effects of ALA as a dietary supplement need to be considered, as all PUFAs are prone to oxidative degradation, producing lipid peroxides and other oxidation metabolites that may be harmful and show adverse health effects [7]. The concentration of lipid oxidation products tends to be higher in produced oils than in the whole foods they derive from or in the body, as treatment processes have to be applied during food production, with thermal, light, and air oxidation being the main pathways of lipid oxidation [8]. An important factor that determines the oxidative stability of a lipid is its fatty acid composition, and ALA oxidation rates are reported to be higher compared to stearic, oleic (OA), linoleic (LA), and gamma linolenic (GLA) acids [9]. Oxidative loss of PUFAs that possess multiple C=C bonds susceptible to oxidation, like ALA (having three double bonds), can be prevented by adding natural or synthetic antioxidants and/or metal-chelating agents to ALA-rich vegetable oils, since naturally present antioxidants only partially inhibit its degradation after exposure to air, light, and heat [10,11]. Micro- and nano-encapsulation represent an innovative approach that led to improvements in the oxidative stability of lipids containing high levels of PUFA [7]. At this moment, the available literature on the association of ALA with adverse effects on health remains inconclusive. Available data indicate that ALA might exert its beneficial preventive effects against cardiovascular diseases and some cancers due to its anti-inflammatory properties [12,13]; however, several studies suggest that ALA intake could be associated with prostate cancer risk and macular degeneration [14].

In addition to the adequate dietary intake of ALA and n-3 polyunsaturated fatty acids (PUFA), the importance of a proper n-6/n-3 PUFA ratio is well-established [4,15], with the high ratio of n-6/n-3 PUFAs being associated with increased synthesis of n-6 pro-inflammatory lipid mediators [12,15]. On the other hand, the modern Western-type dietary pattern has been related to elevated saturated FA and n-6 PUFA intake, together with decreased n-3 PUFA intake [5]. The adequate dietary ratio of n-6/n-3 PUFA is established to be approximately 1:4-5 for normal physiological functions [16], while the typical Western diet has an n-6/n-3 ratio of 10:1 [5,13,15]. Of importance, AL and LA (18:2n-6) are metabolized by the same set of enzymes (desaturases and elongases), and, therefore, the increased LA consumption negatively influences the EPA and DHA synthesis from ALA. 

Metabolic syndrome (MetS) represents a cluster of cardiometabolic risk factors, including abdominal obesity, high triglyceride (TG) levels, low high-density lipoprotein-cholesterol (HDL-C) levels, pre-hypertension or hypertension, impaired glucose regulation, and increased fasting glucose levels. In the available literature, evidence for ALA benefits for MetS is inconclusive. The two recent meta-analyses covering the period from 2005 to 2016 failed to demonstrate a significant association between ALA content in diets or blood and MetS [17,18]. However, favorable effects have been observed in some recent studies [19,20,21,22]. Ngo Njembe et al. (2021) found that consumption of eggs enriched with ALA, docosahexaenoic acid (DHA), rumenic acid, and punicic acid for 3 months affected abdominal obesity, leading to a significant decrease in waist circumference, at the same time showing no effect on other factors of the metabolic syndrome [19]. Egert et al. (2018) have reported that a 6-month hypoenergetic diet supplemented with rapeseed oil rich in ALA induced favorable changes in weight loss [20]. Furthermore, after 6 months of dietary treatment with the same rapeseed oil-enriched hypoenergetic diet, diastolic blood pressure (DP) and TG levels significantly declined compared to an OA-rich diet [22]. At the same time, the ALA-enriched diet induced a significantly higher reduction of human cartilage glycoprotein 39 or chitinase-3-like protein 1 concentration in circulation compared with the intake control diet [23]. In summary, the results of the study on the effects of a hypoenergetic diet enriched in ALA, reported in these three research articles [20,22,23], have revealed that ALA exerts unique and favorable physiological effects during weight loss without changes in polyunsaturated fatty acid profiles in serum and erythrocytes of patients with MetS. Wu et al. (2010) observed that a lifestyle intervention program is effective for the reversion of MetS, including the metabolic variables (weight, waist circumference, serum glucose, total cholesterol (TC), low-density lipoprotein–cholesterol (LDL-C), apolipoprotein (Apo) B, ApoE, and blood pressure) [21]. The consumption of 30 g of walnuts daily together with lifestyle counseling led to a significant reduction in MetS severity (as a mean number of metabolic components). In addition, dietary supplementations with ALA-rich foods, either walnuts or flaxseeds (30 g daily, together with a lifestyle education program), ameliorated central obesity in both intervention groups. Regarding the effects of dietary ALA on cardiometabolic factors, favorable ones on adiposity, TG levels, and hypertension have been generally demonstrated, but the obtained results are still inconsistent [4]. Some studies on ALA supplementation in subjects at a high risk for inflammation, including subjects with overweight/obesity and MetS, showed the systematic anti-inflammatory effects [24,25], but further studies including a larger number of participants are still needed. The existence of a high connection between non-alcoholic fatty liver disease (NAFLD) and MetS, both having visceral obesity as the central feature, has been confirmed in a recent meta-analysis [26]. At this moment, there is some evidence that the Mediterranean diet rich in both plant- and marine n-3 PUFAs could show favorable effects on NAFLD [27]. In summary, according to data in the literature, ALA could show some beneficial effects on MetS features, but current evidence about ALA association with the syndrome is not sufficient due to the inconsistency of obtained results. 

The aim of this narrative review was to summarize the available data in the current literature data on the effects of ALA on MetS. The review focuses on dietary factors and endogenous metabolism of ALA that could lead to the alteration of PUFA profiles and thus influence the synthesis of oxylipins (eicosanoids). A brief overview of factors that influence the efficiency of ALA conversion to its long-chain products EPA and DHA is also given. Moreover, the current knowledge about the possible synergic effects of ALA with EPA and DHA, together with the unique effects of ALA, is briefly reviewed. Finally, data in the recent literature on the associations of changes in desaturase/elongase activity with metabolic health in obesity and MetS are summarized. Additionally, data from recent studies exploring the effects of dietary PUFAs on oxylipins profiles are presented. Furthermore, some new molecular studies exploring the direct ALA-induced changes in oxylipin levels are discussed to clarify the potential favorable effects of dietary ALA in MetS. 

## 2. Literature Search and Study Selection

This narrative review was based on a PubMed electronic database literature search using the following terms (“linolenic acid” OR “metabolic syndrome”) AND “metabolic syndrome” AND (“polyunsaturated fatty acids” OR “n-3 PUFA” OR “flaxseed” OR “walnuts” OR “perilla oil” or “chia”) to identify the relevant publications exploring the association between ALA intake/status and MetS, and/or interventional studies in MetS patients. We also focused on the obtained results of systematic reviews with meta-analyses. The relevant studies on humans, preferably published in the last 10 years in English, were included in this review. The primary list of references that were found to be relevant to this topic was reduced, and the final list of references was approved by the authors. Using as search criteria the terms “linolenic acid” AND “metabolic syndrome”, there were 192 results, and 121 of them were published in the last 10 years. There were 12 results for randomized clinical trials in the last decade. The PubMed search indicated that there were four meta-analyses focusing on the research topic ALA and MetS, but after rechecking, the only reference that met the search criteria was the meta-analysis by Sala-Vila et al. (2022) [4].

## 3. Impact of Dietary α-Linolenic Acid on Metabolic Syndrome

### 3.1. Diagnostic Criteria for Metabolic Syndrome

MetS represents a cluster of interrelated risk factors, including obesity, hyperlipidemia with elevated non-HDL lipoprotein levels, arterial hypertension, and impaired glucose tolerance. According to available data in the literature, MetS affects 20–30% of adults globally, representing a significant health, economic and social problem [28]. The criteria for MetS adopted by the National Cholesterol Education Program (NCEP) Adult Treatment Panel III (ATP III) from 2001 [29] have been commonly used for its assessment worldwide and were adopted with a slight modification by the American Heart Association/National Heart, Lung, and Blood Institute (AHA/NHLBI) in 2005 [30]. According to the NCEP ATPIII/AHA/NHLBI [30], for the diagnosis of MetS in an adult subject of European origin (Europid), at least three of the following criteria need to be met: (1) waist circumference of ≥102 cm in men and ≥88 cm in women, (2) serum TG levels ≥ 150 mg/dL (≥1.70 mmol/L), (3) HDL-C concentration < 40 mg/dL (<1.03 mmol/L) for men and <50 mg/dL (<1.29 mmol/L) for women, (4) fasting blood glucose level of ≥100 mg/dL (≥5.6 mmol/L), and (5) systolic and diastolic arterial blood pressure ≥ 130 and ≥85 mmHg, respectively. There are also other definitions of MetS by the WHO [31], European Group for Study of Insulin Resistance (EGIR) [32], American Association of Clinical Endocrinologists (AACE) [33], and International Diabetes Foundation (IDF) [34], and different cut-off criteria proposed for other races/ethnicities (e.g., Asian populations) [30]. Moreover, the lipid ratios of TG/HDL-C, TC/HDL-C, LDL-C/HDL-C and nonHDL-C/HDL-C were also often used as a marker, instead of single lipid concentrations [35,36,37,38]. Moreover, MetS is associated with additional conditions and diseases [39,40] like chronic inflammation [41], oxidative stress [42], hepatic steatosis (non-alcoholic liver disease NAFLD) [43], cardiovascular diseases (myocardial infarction, atherosclerosis, stroke) [44,45], type 2 diabetes [44,46], impaired kidney function [47], hyperuricemia [48], hypercoagulable/pro-thrombotic states [49], obstructive sleep apnea [50], polycystic ovary syndrome [51], hypogonadism in men [52], lipodystrophies [53], and certain cancer forms (primary affecting GIT—colon, pancreas, and reproductive organs—endometrium, breasts, prostate) [54,55,56].

### 3.2. Effects of Dietary ALA on MetS 

The common non-pharmacological approach in the treatment of MetS implies lifestyle changes focusing on weight control, dietary habits, and exercise practice [57]. In addition to weight loss, the American Diabetic Association (ADA) has recommended a diet with a low content of carbohydrates, saturated and trans fats, and a high fiber amount for improvement of some aspects of MetS, including glucoregulation, arterial hypertension, and hyperlipidemia [30]. At this moment, the evidence for ALA benefits for MetS is inconclusive [4,58], and the available literature concerning ALA association with MetS in the period up to 2016 is summarized in two recent meta-analyses [17,18]. A meta-analysis of cross-sectional and case–control studies that examined the association between biomarkers of n-3 PUFA status and MetS prevalence [17] revealed that higher content of total n-3 PUFA in circulation was related to lower MetS risk. However, no significant relationship has been found for the ALA levels. Similarly, the meta-analysis by Jang and Park [18], including mostly cross-sectional studies using ALA status as a biomarker of dietary exposure, showed no significant association of ALA status with MetS prevalence. Sala-Vila et al. (2022) reported several recent studies that were not included in these meta-analyses and evaluated the potential impact of ALA status on MetS prevalence [4]. Furthermore, several relevant publications appeared in our PubMed search, and the obtained results of all of these references are summarized in Table 1 [19,21,22,59,60,61,62,63,64,65] (Table 1). The content of ALA in erythrocytes was associated with higher MetS prevalence in two recent cross-sectional studies [60,61]. At the same time, Muzsik et al. (2020) did not observe a significant difference in ALA content in erythrocytes when comparing postmenopausal women with and without MetS [65]. Finally, low adipose tissue ALA content was found to be associated with MetS incidence in a cross-sectional study by Flannagan et al. (2018) [59]. 

ALA can be at least partly converted to corresponding very long-chain n-3 PUFA, including EPA, DPA, and much less DHA [66,67,68,69,70,71,72,73,74], with the delta-6-desaturase (D6D) being the rate-limiting factor for the conversion of ALA to EPA. In MetS, increased D6D activity with decreased delta-5-desaturase (D5D) activity was observed in the serum and erythrocytes [75]. So, it might be important to take into consideration all ALA-induced changes in serum/circulation/tissue PUFA profiles (both in n-3 and n-6), as well as the changes in ALA content when analyzing ALA benefits. It is estimated that the rate of conversion to EPA and DHA is very low, about 5–12% for EPA and less than 1% for DHA, with infants and women having a higher ability to convert ALA to EPA and DHA. The estimated mean net conversation rate of ALA to EPA was 21% in women compared to 8% in men, and to DHA was 9% in women and 0% in men [3,76]. Recently, Drobner et al. (2023) reported that ALA supplementation led to a significant increase in ALA, stearidonic acid (STE, 20:4n-3), and EPA content in the erythrocytes [74]. At the same time, DHA, Omega-3 Index (sum of EPA and DHA percentage in erythrocytes), di-homo-gamma-linolenic acid (DHGLA, 20:3n-6), arachidonic acid (AA, 20:4n-6), and docosatetraenoic acid (DTA, 22:4n-6) contents were decreased. Finally, n-3 DPA and n-6 LA levels were increased only in some of the treatment groups. ALA also could be immediately oxidized, and thus, no changes in ALA content would be observed. There is a high rate of ALA oxidation: it is estimated that 60–85% of consumed ALA is oxidized [3]. In accordance, in a study by Egert et al. (2018), consumption of a hypoenergetic diet rich in ALA for 26 weeks did not lead to an increase in ALA content in MetS women [20]. For these reasons, valuable information about dietary ALA effects on MetS could be obtained from ALA intervention trials and studies in which the total dietary intake of ALA was analyzed. So far, in a large Spanish cohort, the authors have not found a significant association between dietary intake of ALA and the MetS criteria in 6560 older MetS patients [62]. Of note, the authors did not evaluate the relation between ALA intake and MetS prevalence, but the influence of FA intake on cardiometabolic risk factors in MetS subjects [62]. Finally, a few recent randomized control trials (RCTs) have analyzed the effects of ALA-enriched diets, usually flaxseed or walnut supplemented, on MetS components, showing both favorable [20,21,22] and no effects on MetS risk [63]. Summarizing the results of walnut intake intervention trials, Mates et al. (2022) revealed favorable effects of walnut consumption on MetS hypertriglyceridemia and some inflammation markers [25]. In a recent interventional study, ALA supplementation in combination with DHA, rumenic acid (cis-9, trans-11 18:2), and punicic acid (cis-9, trans-11, cis-13 18:3) for 3 months led to favorable changes in adiposity but did not show an effect on the other components of MetS in subjects at risk of developing MetS [19]. In the study of Bellian et al. (2022), ALA supplementation with camelina oil that contained about 1.5 g of ALA in a daily dose did not improve vascular function in MetS subjects after 6 months of the treatment [64]. At the same time, it induced an adverse effect on insulin sensitivity that seemed to be related to gonadic acid present in relatively high amounts in camelina oil and needs to be further explored [64]. Further studies aimed determining PUFAs and oxylipins profiles in a larger number of participants could reveal if dietary ALA shows beneficial effects in MetS.

### 3.3. Effects of ALA on Cardiometabolic Factors Associated with MetS

Regarding the impact of dietary ALA on cardiometabolic factors, favorable effects on adiposity, TG levels, and hypertension have been observed in some studies, but not confirmed in all (Figure 1). 

#### 3.3.1. Effect of ALA on Abdominal Obesity

Long-term body-weight management could be influenced by the type of fat consumed. The effects of dietary intake of ALA-rich sources have been recently summarized in meta-analyses, showing no significant effects for walnuts [77], but favorable effects for flaxseeds [78].

After ingestion, about 15–35% of ALA is immediately beta-oxidized within the first hours, while within 7 days, about 70% is oxidized to CO_2_, with a part of the oxidized carbon being recycled and used for de novo synthesis of FAs and cholesterol; thus, the percentage of ALA in adipose tissue TG is only about 1% of total FAs [76]. A hypoenergetic diet rich in ALA (3.4 g/day) in subjects with MetS did not increase its content in serum phospholipids after 26 weeks of consumption compared to the control diet (0.9 g ALA/day) [20]. So, the beta-oxidation of dietary ALA seems to be a primary metabolic fate in humans, and thus, ALA consumption could influence adiposity and abdominal obesity. ALA has a higher preference for beta-oxidation in mitochondria; compared to LA, OA, and AA, LA does not influence the oxidation of ALA, and compared to LA, ALA has a lower tendency to be stored in TG, both in hepatocytes and abdominal adipose tissue [79,80,81,82]. The higher ratio of ALA/LA in the diet was associated with increased FA oxidation and decreased FA accumulation pathways in the rodent liver, which could lead to a decreased level of adiposity [79,83,84].

ALA and its n-3 elongation products (as well as other FAs) are ligands for peroxisome proliferator-activated receptors (PPARs) and sterol response element binding protein-1c (SREBP-1c) [76]. PUFAs are very potent PPAR ligands, but even more potent and more specific PPAR ligands are the lipoxins (eicosanoids) and endocannabinoids, which derive from the PUFA transformation. Both DHGLA and EPA compete with AA for the lipoxin and endocannabinoid (EC) synthesis, and they can influence the synthesis of products that have the opposite effect of AA products, not only simply diminish AA transformation [85,86,87]. SREBP-1c is the key enzyme in lipogenesis and lipid accumulation, while some PPARs are obesogenic (e.g., PPAR-gamma, which stimulates lipogenesis) and others have an anti-obesogenic effect (e.g., PPAR-alpha and PPAR-beta/delta, which increase lipid oxidation). ALA stimulates the expression of PPAR-alpha [79]. Similarly, ECs derived form AA are shown to stimulate appetite, increase lipogenesis and lipid accumulation, and decrease lipid oxidation, by increasing SREBP-1c, carbohydrate regulatory element binding protein (ChREBP), liver X receptors (LXRs), and PPAR-gamma and decreasing AMP-activated protein kinase (AMPK) and PPAR-alpha expression and activity in the liver, muscle, and adipose tissue [75,88]. Moreover, ALA can influence the adipocyte fluidity and the levels of basal and catecholamine-stimulated lipolysis, as well as the antilipolytic effect of insulin [89].

#### 3.3.2. Effects of ALA on Hypertriglyceridemia and Low HDL-Cholesterol

Most studies have revealed that dietary exposure to n-3 PUFA results in favorable changes in lipid profiles. Regarding ALA effects, the authors of a meta-analysis [89] that summarized the effects of flaxseed dietary supplementation on lipid profiles reported modest reductions in TC and LDL-C. At the same time, they found no significant changes in TG and HDL-C levels. A meta-analysis by Yue et al. [90] demonstrated that ALA intake leads to a reduction in the TC, LDL-C, and TG levels, compared to the control. In a meta-analysis by Yin et al. (2023), the authors observed that ALA induced a significant reduction in TG levels, increases in LDL-C levels, and no significant changes in TC and HDL-C levels in obese and overweight individuals [24]. The subgroup analysis also revealed the importance of intervention duration and ALA dosage, showing that a dose of ≥3 g/day and a duration of ≥12 weeks induced more prominent effects. The effects of walnut interventions on lipid profile were examined in a recent meta-analysis [91]. The obtained results indicated that their intake significantly decreased TC, LDL-C, and TG levels, compared to a control diet [91]. A meta-analysis by Mates et al. (2022) examined the effects of walnut intake on MetS and inflammation in middle-aged and older adults [25]. Regarding lipid profiles, walnut-enriched diets significantly decreased TC, LDL-C and TG levels [25]. In a recent meta-analysis by Wang et al. (2023), the ALA/LA ratio increased by supplementation in adults was associated with a significant reduction in plasma TC, LDL-C, and TG concentrations, but no changes in HDL-C concentrations [92]. 

#### 3.3.3. Effect of ALA on Hypertension 

Hypertension is the most frequent pathophysiological condition globally, with prevalence reaching 35% in the adult population [93]. Pre-hypertension/hypertension is an important element of MetS that was found to be present in 90% of MetS individuals [94]. A meta-analysis examining the effects of interventional studies using flaxseed, as a relevant source of ALA, on blood pressure outcomes showed that both systolic and diastolic blood pressure were significantly reduced by the intervention, and the effect was greater if it lasted ≥12 weeks [95]. Yin et al. (2023) have demonstrated that ALA supplementation induced a significant decrease in systolic pressure with no change in diastolic pressure in their meta-analysis of available data on obese and overweight subjects [24]. However, several studies reported no effects of ALA supplementation on blood pressure [96,97].
Figure 1The potential favorable effects of dietary α-linolenic acid on main components of metabolic syndrome on abdominal obesity [78]; hypertension [95]; dyslipidemia [25,90,91,95]; and prediabetes, diabetes in human studies; none of the effects was confirmed in all studies; TG, triglycerides; LDL, low-density cholesterol; HDL, high-density cholesterol; HbA1c, glycated hemoglobin A1c; HOMA-IR, Homeostatic Model Assessment for Insulin Resistance; ? meaning is that potential beneficial effects are still questionable, unknown.
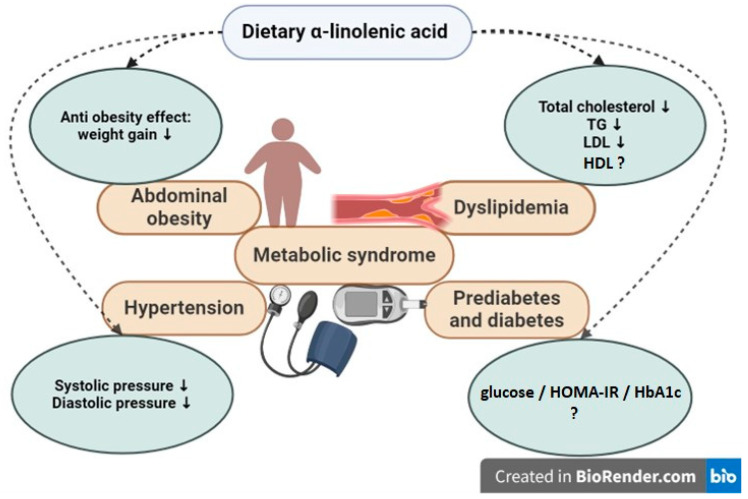


#### 3.3.4. Effect of ALA on Impaired Glucoregulation

One component of MetS is pre-diabetes or diabetes. Pre-diabetes is generally defined by either impaired fasting glucose or impaired glucose tolerance (ADA) [98]. The two metabolic changes that lead to pre-diabetes and type 2 diabetes developments are insulin resistance and deficient insulin secretion [99]. Bianchi et al. (2011) observed that about three-fourths of MetS patients had combined impaired fasting glucose and impaired glucose tolerance [100]. 

Numerous available data on cultured cells or animals indicated that ALA could improve insulin sensitivity or glucoregulation through various mechanisms [88,101,102,103,104]. However, in humans, according to data in the currently available literature, intake of ALA seems to have little or no effect on type 2 diabetes (T2D) risk or parameters of glucose homeostasis, including fasting glucose, fasting insulin, glycated hemoglobin A1c and Homeostatic Model Assessment for Insulin Resistance (HOMA-IR) approximative index of insulin resistance [105,106]. Recently, a lack of significant effects of walnut-enriched diets on glycemic markers, including fasting blood glucose, fasting insulin, glycated hemoglobin, and HOMA-IR value, was described in a meta-analysis by Mates et al. (2022) [25]. A recent meta-analysis by Chen et al. on cardiometabolic risk factors also showed no effect on fasting glucose [107].

In animals, a positive effect on the adipose tissue insulin signaling pathway was shown, including insulin-receptor signaling and abundance of the insulin receptor, protein-kinase-A (PKA) signaling, and phosphorylation of protein kinase B (Akt), ERK/MAPK and p38 MAPK signaling, AMPK signaling, calcium signaling pathways, and lipolysis [101]. Additionally, ALA decreased plasma and adipose tissue concentrations of ECs, which are products of the metabolism of AA, OA, and palmitoleic (POA, 16:1n-7), by the activity of N-acyl phosphatidylethanolamine phospholipase D (NAPE-PLD) and diacylglycerol (DAG)-lipase [108]. The activation of the EC system (particularly endocannabinoid CB1 receptor) is associated with the induction of insulin resistance in adipose tissue, liver, and muscle, and the overactivation of ECS was shown in obesity (particularly abdominal obesity) and MetS [87]. ALA and its desaturation and elongation n-3 products negatively influence the activity of genes involved in the synthesis of ECs, increase their degradation by the enzymes fatty acid amide hydrolase (FAAH) and monoacylglycerol (MAG)-lipase, and decrease expression of the endocannabinoid receptors CB1 [108]. ALA was shown to increase glucagon-like peptide-1 (GLP-1) secretion, thus stimulating insulin [104]. ALA can influence the content of trans FAs in membranes, oxidative stress, fluidity of membranes, and adipocyte basal and catecholamines-stimulated lipolysis, as well as anti-lipolysis, which can reduce the negative effect of lipotoxicity on insulin sensitivity [88,109].

### 3.4. Effects of ALA on Systematic Inflammation and Non-Alcoholic Liver Disease (NAFLD), Which Are Tightly Associated with MetS

#### 3.4.1. Effects of ALA on Systematic Inflammation

Obesity is often defined as a persistent low-grade inflammation state. The markers of inflammation related to obesity include profiles of adipose tissue-derived adipokines (namely leptin, adiponectin, and resistin), serum levels of liver-synthesized C-reactive protein (CRP), and concentrations of some inflammatory cytokines: interleukin (IL)-6, tumor necrosis factor (TNF)-α, IL-1β, produced by immune cells [110,111]. Obesity, diabetes, and related disorders induce monocyte recruitment into the tissues (mainly „transformed” white adipose tissue) and resident macrophage proliferation, with a switch toward a more pro-inflammatory M1-like state [112]. Rohm et al. (2022). suggested that the two key signaling pathways that are up-regulated in obesity-induced inflammation are those leading to nuclear factor κB (NF-κB) translocation to the nucleus and NLRP3 “inflammasome” activation [113]. Macrophages that infiltrate adipose tissue, due to the activation of NF-kB, produce high levels of TNF-α [114]. At the same time, the activation of a multimeric protein complex NLRP3 “inflammasome” by cell nutrients like glucose and free FAs leads to increased production of IL-1β, by an increased caspase-1 action [115].

The beneficial role of dietary n-3 PUFAs on inflammation markers has generally been reported in subjects with high inflammatory states, like obesity [24,25]. However, the findings on beneficial effects of ALA on markers of inflammation are not consistent, as the favorable effects have not been shown in some RCTs [4]. Recently, in a WAHA trial with many participants, some beneficial effects of walnut consumption for 2 years (as 15% of energy intake) were reported concerning the following markers of inflammation: granulocyte-monocyte colony-stimulating factor, interferon (IFN)-γ, IL-1β, IL-6, TNF-α, and E-selectin in older subjects [116]. Several narrative reviews that have described the effects of n-3 PUFAs on the immune system and cells [117,118,119] focused on studies that explored the direct in vitro effects of n-3 PUFAs on immune cells [119]. Considering effects on macrophages, the authors reported that n-3 PUFAs influence the production and secretion of cytokines, decrease reactive oxygen species (ROS) and NO production, and increase polarization toward the M2 phenotype and phagocytotic capacity of macrophages. At the same time, the n-3 PUFAs have been found to influence activation, migration, phagocytosis, infiltration, and IgM production in other types of immune cells. Their effects on immune cells are known to be achieved by multiple mechanisms, including alteration of membrane properties, interaction with the surface receptor, binding to G-protein coupled receptors, and nuclear factors [119,120,121]. Moreover, Rodaway et al. (2023) examined the effects of ALA on the immune cell transcriptome and observed that ALA and DHA show distinct effects on the expression of genes involved in cholesterol metabolism in THP-1 monocytes of obese humans [122]. Furthermore, the studies exploring ALA effects in animals and in vitro models showed that ALA reduces TNF-α production in macrophages [123,124]. ALA effects on the production of pro-inflammatory molecules in macrophages are mediated by PPAR-γ activation and through the blocking of NF-kB [124,125]. Finally, the ALA-derived 13-hydroperoxy-octadecatrienoic acids (13-HpOTrE) and 13-hydroxy-octadecatrienoic acids (13-HOTrE) reduced the production of pro-inflammatory cytokines (IL-1β, TNF-α) in peritoneal macrophages stimulated by lipopolysaccharide (LPS), leading to PPAR-γ dependent inactivation of NLRP3 “inflammasome” [126]. The EPA, DHA, and their oxygenated metabolites are strong natural ligands for PPARs [127]. So, dietary anti-inflammatory effects could be achieved after conversion of ALA to EPA and DHA, but there is accumulating scientific evidence that they could be mediated by ALA directly, through derived octadecanoic oxilipins (9-HOTrE and 13-HOTrE), as recently reported by Cambiaggi et al. (2023) [128]. 

Collectively, studies examining the effects of dietary ALA in the context of obesity indicate that it may show favorable effects on obesity-associated inflammation that is likely mediated (at least partly) by oxilipin metabolites [122,128].

#### 3.4.2. Effects of ALA on NAFLD

In recent years, the increasing prevalence of NAFLD has been noticed due to lifestyle changes, including dietary habits. The systematic review by Zohara et al. (2023), demonstrated the existence of a strong connection between NAFLD and MetS, with visceral obesity as the central feature [26]. The limited treatment options for NAFLD, including dietary modifications, regular physical activity, and gradual weight loss, make its management challenging. Maintaining an optimal dietary n-6/n-3 PUFA ratio has been identified as a prevention and management measure [27]. In a recently conducted RCT, six-month treatment with n-3 PUFAs (daily dose 0.945 g; 64% ALA, 21% EPA, and 16% DHA) in patients with non-alcoholic steatohepatitis (NASH) led to favorable changes in their proteomic and lipidomic markers of lipogenesis, reduction of endoplasmic reticulum stress (ERS) and improvement in mitochondrial function [129]. The calorie-restricted weight loss trial has revealed that the Mediterranean diet type, rich in OA and n-3 PUFA (nuts-derived ALA and marine-derived EPA and DHA) induced a greater decrease in markers of NAFLD [27] than a low-fat diet, indicating that beneficial effects were beyond the favorable changes induced only by visceral fat loss. A study by Ristic Medic et al. (2020) showed that FA profiles were favorably modified after calorie-restricted Mediterranean and low-fat diets, both inducing a significant decrease in the n-6/n-3 ratio [27]. However, participants on the Mediterranean calorie-restricted diet had higher levels of monounsaturated FAs and DHA and lower levels of saturated FAs in serum phospholipids, together with more pronounced, favorable changes in HDL-C, TG, and the TG/HDL ratio compared to participants on the low-fat diet. Finally, the green Mediterranean diet, enriched with green plant-based proteins/polyphenols due to dietary intake of high amounts of Mankai, green tea, and walnuts, and restricted in red/processed meat, has been shown to reduce twice the hepatic fat compared to some other nutritional strategies [130].

ALA was shown to influence not only the oxidation or storage of fat in the liver, by increasing the expression or activity of the enzymes and nuclear factors (including PPAR-alpha) [79,83,84], but also influence de novo lipogenesis by inhibiting delta-9-desaturase (D9D, SDC-1) in the liver, adipose tissue, and muscle [131,132,133], a rate-limiting enzyme with activity shown to be increased in MetS and NAFLD [75]. ALA as well as OA, LA, EPA, and DHA reduce the D9D activity by inhibiting nuclear factors (including SREBP-1c) that are involved in its transcription and activation [131]. By those multiple pathways, ALA decreases hepatic lipid accumulation. Moreover, ALA can increase the antioxidant defense and decrease oxidative stress, ERS, and production of inflammation markers in the liver [102,109].

## 4. Endogenous Metabolic Transformation of ALA to Its Long-Chain Products and the Impact of ALA on Oxylipins

The very important scientific question is whether ALA transformation to EPA and DHA is of critical importance for its potential beneficial effects on MetS, both in prevention and treatment. Currently, data on the effects of ALA on MetS are inconsistent and inconclusive. So, further studies estimating dietary intake and status (determining FA profiles) as well as synthesis of lipid mediators derived from ALA may be needed. 

### 4.1. Metabolism of ALA to Long-Chain n-3 PUFAs Metabolites: Desaturation/Elongation Pathway 

Two series of PUFAs, n-3 and n-6, are endogenously synthesized from parent fatty acids ALA and LA [134]. These PUFAs are essential for mammals, as they lack desaturases that could insert the first *cis* double bond between the third and the fourth carbon atom and the sixth and seventh from the methyl end of the FA [135]. ALA and LA are metabolized by an alternating series of elongation and desaturation reactions to major product AA from LA and EPA and DHA from ALA sharing the same set of enzymes for both pathways [136] (bio-conversion pathway for ALA is presented in Figure 2).

The increased dietary intake of ALA decreases the production of metabolites of the n-6 pathway; however, the conversion of ALA to EPA and its further conversion to DHA are not efficient [134,136]. In humans, ALA conversion to EPA is only limited, and its further conversion to DHA rather marginal [66,67,68,69,70,71,72,73,74] (as shown in Table 2). Fish oil consumption significantly increases plasma and erythrocyte content of DHA, DPA n-3, DHA, and total n-3 PUFA [137], whereas ALA supplementation usually raises only ALA and EPA levels, as well as DPA n-3 in some studies [66,67,68,69,70,71,72,73,74] (Table 2). Nevertheless, the DHA content was maintained at the same level in different lipid pools in the body after ALA supplementation in most studies [66,67,68,69,70,71,72,73,74] (Table 2). D6D is involved in both the conversion of ALA to STE and DPA n-3 to DHA, and this enzyme may not be available in adequate amounts for both reactions. D6D has a higher affinity to ALA than to tetracosapentaenoic acid of the n-3 family (TPA, 24:5n-3) which is a D6D precursor during DHA synthesis, so DHA production is decreased and limited [138]. In addition, D6D is also required for conversion of n-6 PUFA, which could explain the decrease in AA and DPA n-6, together with the increase in LA after supplementation with ALA due to higher affinity of D6D to n-3 compared to n-6 PUFA [135]. Moreover, the adequate duration of treatment is of critical importance for the induction of changes in PUFA profiles in erythrocytes and tissues since no changes were found in some short-term dietary intervention studies [139]. Finally, in a study by Egert et al. (2018), the consumption of a hypoenergetic diet high in ALA (3.4 g/d), compared to a control diet low in ALA (0.9 g/d), did not increase ALA, EPA, DPA n-3, or DHA levels in overweight or obese subjects with MetS, probably due to high rates of immediate ALA oxidation in hypoenergetic catabolic states [20].

### 4.2. Can the Efficiency of the ALA Conversion to Long-Chain n-3PUFA Metabolites Be Enhanced by Dietary Factors? 

An increased amount of EPA and DHA in erythrocytes has been associated with reduced risk for MetS in previous studies, but the effect of ALA has not been clearly confirmed [17]. The cardioprotective effect of plant-derived ALA seems to be weaker compared to those of marine-derived n-3 PUFA EPA and DHA [2,4,58,140,141]. However, as the dietary intake of EPA and DHA is generally low worldwide [142], the important scientific question that needs to be addressed is whether ALA conversion to long-chain n-3 PUFAs can be significantly influenced by other dietary components and whether it can be favorably modified by changing dietary habits. The genetic influence on the activity of the conversion enzymes is well established, and the increased risk of MetS was associated with some single nucleotide polymorphisms (SNPs) of the genes involved in the synthesis of long-chain PUFAs [76].

Numerous studies in animals and humans have shown the existence of sex differences considering the efficiency of ALA conversion to EPA, DPA n-3, and DHA due to estrogen-mediated upregulation of D5D and D6D and lower rates of β-oxidation in females [76,143,144]. However, the actual percentage of ALA conversion cannot be easily determined, as ALA, EPA, DPA n-3 and DHA also could be released from, synthesized in, or stored in different lipid pools, or further metabolized. Tracer studies with stable isotopes have been conducted to overcome these limitations [65,66,73,143]. Only a small proportion of the ingested ALA in plasma was found to be converted to EPA (6–8% in men and 21% in women), DPA n-3 (4–8% in men and 6% in women), and DHA (0–4% in men and 9% in women) [66]. In contrast to data available in the literature, Drobner et al. (2023) did not find that conversion of ALA to EPA was more efficient in women, possibly because the study’s female participants were in menopause [74]. However, the conversion rates were found to be much lower in another study (0.2% for EPA, 0.13% for DPA n-3, and 0.05% for DHA) [73]. In the conversion process, the activity of D6D to produce STE seems to be the rate-limiting step of ALA to EPA conversion, but also the accumulation of TPA, as a product of elongase 2, can also exert negative feedback on the elongase 2 activity, indicating that elongase 2 is the rate-limiting step for the EPA to DHA conversion [76]. 

According to data in the literature, numerous dietary factors can influence the efficiency of the metabolic transformation of ALA to its long-chain products. The effects of the dietary intake and status of marine-derived EPA and DHA on the conversion of ALA have been described by Welch et al. (2010), who compared the elongation/desaturation of ALA in non-fish-eaters (low EPA and DHA intake/status) to fish-eaters (high EPA and DHA intake/status) [144]. Their study confirmed that exogenous EPA and DHA lower the conversion of ALA to long-chain PUFAs. Since the same set of enzymes is involved in the metabolism of both n-6 and n-3 PUFAs, LA intake has been also shown to decrease the conversion of ALA to EPA, DPA n-3, and DHA, due to their competition for the same desaturases and elongases [143,145,146,147,148]. It was shown that desaturases had a higher preference toward n-3 PUFAs than n-6 PUFAs [76].

In a recent study, Drobner et al. (2023) evaluated the influence of several factors, including EPA baseline status, dietary intake of LA and milk fat intake, on the metabolic transformation of ALA into its long-chain products [74]. The participants in this RCT (n = 134) were assigned to one of four diets (high in LA (HLA); low in LA (LLA); high in milk fat (MF); and control (Western diet)), each enriched with flaxseed oil in a dose that provided 13–16 g of ALA daily. The ALA supplementation led to a significant increase in ALA, STE, and EPA content in erythrocytes in all four groups, while DPA n-3 only increased in the MF group. At the same time, DHA, Omega-3 Index, DHGLA, AA, and DTA levels dropped, whereas LA content was increased in the HLA, LLA, and control groups. The milk fat-specific fatty acids (short-chain, medium-chain, branched-chain, and conjugated fatty acids) have already been found to influence the metabolic transformation of ALA into EPA, DPA n-3, and DHA in previous studies [148,149,150,151]. The high EPA status in erythrocytes at baseline (>0.9%) and high LA status were found to attenuate the rate of ALA conversion to EPA [74]. 

Dietary FAs can influence the long-chain PUFA biosynthesis pathway at the level of transcription of D5D (encoded by *FADS1)* and D6D (encoded by *FADS2*), and elongases, by the activation of PPARs and SREBP-1c, and FAs are ligands for those nuclear factors (reference Baker/Cadler). High EPA and DHA intake in fish oil was found to suppress the expression of D6D and elongase 5 [76].

In addition to the dietary intake/status of different FAs, some micronutrients are found to influence the activity of D5D and D6D desaturases, including folate, vitamin A, zinc, copper, and iron. Rat studies indicate folate in combination with inadequate B12 (low or absent) leads to a decrease in D6D activity and could influence the activity of D5D as well as expression of its *FADS1* gene [152,153,154]. Homocysteine (which is increased when vitamin B12 and folate are low) can inhibit the expression of D6D by modifying DNA methylation [73]. Dietary exposure to vitamin A raised estimated indices of D5D and D6D activities, but decreased *FADS1* expression [153]. D5D and D6D are both zinc-dependent enzymes whose deficiency led to reduced activities of these enzymes as well as expression of *FADS1* and *FADS2* [153,155]. Even though the influence of micronutrients on desaturase activities and expression of their genes is usually studied in animal models, there are several studies focusing on zinc effects even in humans [156,157]. Similarly to zinc, iron is a co-factor of desaturases and has been shown to alter long-chain PUFA synthesis in animal models and humans [153]. Moreover, dietary polyphenols, plant-based bioactive molecules, can also affect the desaturation/elongation of n-6 and n-3 PUFAs by influencing the expression and activities of desaturases and elongases [153,158].

In summary, in addition to genetic factors, sex, and hormones, some dietary components like different types of FAs, micronutrients (vitamins and minerals), and polyphenols are known to influence both desaturases’ activities and expressions. The combination of some nutrients with ALA could raise the efficiency of the conversion of ALA into its long-chain n-3 PUFA products and could be explored in patients with MetS as a dietary therapy option. Furthermore, the reduction of the dietary intake of LA, leading to a lower n-6/n-3 ratio than that of Western-type diet intake, could improve the efficiency of ALA conversion to EPA and consequently show a more pronounced favorable effect on MetS. 

### 4.3. Does ALA Show Some Unique Effects Different from EPA and DHA?

Currently, there is not much available scientific data concerning direct comparisons the favorable effects of plant-derived ALA and fish oil-derived EPA and/or DHA in subjects with obesity and MetS. However, there is no doubt that n-3 PUFA properties are not equal [159]; for example, supplementation with DHA increases LDL-C, while EPA does not [4]. In a recent study, Pauls et al. (2021) observed that supplementation with ALA- and DHA-rich oils each had different effects on markers of metabolic health in obese women [160]. The supplementation with 4 g/day DHA (plus 1 g/day of EPA) significantly increased concentrations of EPA- and DHA-derived oxylipins, favorably altered HDL-C and TG levels, and led to an increase in plasma adiponectin, but induced no changes in systemic inflammatory markers. On the contrary, ALA intake (4 g/day, 4 weeks) induced marked change in monocyte bioenergetics, resulting in decreased production of reactive oxygen species (ROS) and suppression of pro-inflammation. The interchangeable plant-derived ALA and marine-derived DHA could exert full benefits when present together in a diet due to the existence of complementary mechanisms of action [141,160,161].

Erythrocyte fatty acids are biomarkers of dietary exposure and reflect n-6 and n-3 PUFA intake in the last 90 days, but also their metabolic conversion. The purpose of a study by Liu et al. (2022) was to compare the effects of different dietary sources of n-3 PUFA (plant-based ALA and marine-derived EPA and DHA) on glucose and lipid metabolism in T2D patients [162]. In addition to the well-known effects of n-3 PUFAs (especially EPA and DHA) on lipid markers, they also could improve insulin resistance by inhibiting pro-inflammatory cytokines and promoting the resolution of inflammation. Data in the literature on the impact of n-3 PUFAs on glucose regulation and resistance are inconsistent, but the trial of Liu et al. [162] demonstrated that perilla oil intervention more effectively reduces fasting glucose and HbA1c levels compared to fish oil. As perilla oil is a common plant-based source rich in ALA, the obtained results suggest the potential ability of this FA to improve glycemic control. These findings are in accordance with the results of some ALA supplementation studies [163,164]. Erythrocytes’ ALA, DPA n-6, DPA n-3, n-6/n-3 ratio, and AA/EPA levels were higher in the perilla oil group compared to the fish oil group and in those subjects that mix flaxseed oil with fish oil, whereas EPA, total n-3 PUFAs, and Omega-3 index were higher in the fish oil group compared to the perilla oil group. Comparing groups, TG levels decreased only after fish oil treatment and glucose levels after perilla oil treatment, along with different changes in erythrocyte PUFA composition [163]. This study confirms that marine- and plant-based n-3 PUFAs exhibit different effects considering lipid and glucose metabolism. 

### 4.4. The Direct Synthesis of Lipid Mediators (Oxylipins) from ALA

The lipoxygenases (LOX), cyclooxygenases (COX), and cytochrome P450 oxidase/epoxygenases (CYP) generate a series of lipid mediators—eicosanoids from AA, EPA, and DHGLA. LOX and CYP are also involved in the production of E-resolvins from EPA, while LOX converts DHA to D-resolvins, maresins, and protectins. A comprehensive review concerning the production of lipid mediators has recently been published [165]. An important issue is that the relative abundances of C20 PUFAs at least partly determine the production of lipid mediators that are involved in inflammation processes [139]. The AA-derived mediators are mainly pro-inflammatory, while those that derive from n-3 PUFAs show the opposite effect [119]. Thus, the n-6/n-3 ratio in the diet is an important factor influencing the balance between the generation of pro-inflammatory and anti-inflammatory oxylipin molecules. The recommended n-6/n-3 ratio is 4–5 times lower than in the current typical Western diet [5,13,15,27]. In addition, endogenous elongation/desaturation is a key factor that influences the AA/EPA, AA/DHA, and AA/(EPA+DHA) ratios [1,2]. Nowadays, many of the most prevalent pathological conditions are found to be associated with low-grade persistent systematic inflammation, including MetS [166].

CYPs can convert both LA and ALA to 18C oxylipins that are easily altered by diet [167,168]. Recently published data by Cofan et al. (2023) indicate that consumption of a diet enriched with walnuts as 15% energy for 2 years in healthy older males and females, compared to a control diet, led to statistically significant increases in the ALA-directly derived oxylipins 9-HOTrE, 13-HOTrE, and 12,13-epoxy-octadienoicacid (12,13-EpODE), and in the EPA-derived oxylipins 14,15-dihydoxy-eicosatetraenoic acid (14,15-diHETE), while reducing the AA-derived oxylipins 5-hydroxytetraenoic acid (5-HETE), 19-hydroxytetraenoic acid (19-HETE), and 5,6-dihydroxy-trienoic acid (5,6-diHETrE) [169]. The ALA-derived lipid mediators seem to play an important role in the inflammation associated with metabolic syndrome and cancer, and their role in cardiovascular health has been recently reviewed [128].

Considering the mechanisms of action, LA is the most abundant PUFA in typical human diets, and its CYP products tend to be pro-inflammatory, activating nuclear factor (NF)κB, which increases oxidative stress [170,171] or changes lipid metabolism through the activation of PPAR-gamma [127]. The ALA-derived oxylipins seem to have the opposite effect compared with LA-derived oxylipins and play a role in reducing inflammation [170]. The mechanism of their action is through inactivation of NOD-like receptor protein 3 (NLRP3) inflammasome [127]. Upon binding and activation of n-3 PUFA to G120 (also called free fatty acid receptor 4), highly expressed in adipocytes, endothelial cells, and macrophages, a complex with β-arrestin-2 can be formed that inhibits NLRP3-dependent inflammation [172,173]. A study by Kumar et al. [127] in cultured macrophage cells and mice showed that LOX-15 metabolites of ALA (13-HpOTrE and 13-HOTrE) downregulate NLRP3 inflammasome and iNOS expression in LPS-stimulated macrophage cells through the upregulation of COX-2, increased production of prostaglandins PGD2 and PGE2, and subsequent PPAR-γ activation, leading to an inhibition of NF-κB translocation, a decrease in the production of nitric oxide, ROS, inflammatory cytokines IL-1β, IL-18 and TNF-α, and an increase in the production of anti-inflammatory IL-10 and apoptosis. There is a well-known link between inflammation (particularly the NF-κB pathway, IL-1β, and TNF-α) and insulin resistance development, which is the key feature in MetS [113]. Although PGE2 is generally considered to be pro-inflammatory, in macrophages it leads to a shift from the M1 to M2 subtypes that have anti-inflammatory properties, while PGD2, through its conversion to PGJ2 and its product 15-deoxy-Δ 12,14 PGJ2 (15d-PGJ2), a potent ligand for PPAR-γ, exhibits anti-inflammatory effects and increases insulin sensitivity [113]. The downregulation of iNOS and NLRP3 expression, as well as decreased IL-1β production but increased IL-10 production, was shown also in the macrophages, liver, and blood of mice in vivo injected with 13-HpOTrE and 13-HOTrE [127]. A recent study has shown that ALA supplementation in mice fed a high-fat diet (HFD) leads to an equally increased production of both ALA-derived and EPA-derived 5-LOX and 15-LOX oxylipins; therefore, the positive effects can be both a direct consequence of ALA and indirect, through its conversion to EPA, which is important to distinguish [174]. Unfortunately, there are no studies on the effects of the oxylipins directly derived from ALA on the metabolic features connected with MetS, and further studies in this direction are needed. In obese animals fed an HFD, decreased levels of both ALA- and EPA-derived oxylipins were shown in adipose tissue [175]. Recently, Fisk et al. (2022) demonstrated that the levels of these oxylipins and the expression of different genes of proteins involved in oxylipin synthesis in subcutaneous white adipose tissue are altered in obese subjects [176]. Moreover, the effects of exposure to fish oil n-3 PUFAs on adipose tissue inflammation were found to be reduced in obese subjects [176]. The beneficial effects of EPA and DHA in obesity are well-documented, but a more personalized approach concerning their dietary intake may be needed. Finally, Grabss et al. (2021) demonstrated on a molecular level that the health effects of dietary ALA and DHA could be mediated at least partly by their different effects on oxylipins [177]. A greater effect of DHA supplementation, compared with a similar dose of ALA supplementation, was observed on both n-3 and n-6 PUFA-derived oxylipins in young, healthy adults [177].

### 4.5. Endogenous Metabolism of PUFAs in Metabolic Syndrome Subjects

In the last 10 years, the connection between FA compositions in various body compartments with MetS and cardiometabolic risk factors in overweight and obese subjects has been evaluated in several studies [4,19,59,60,61,66]. The altered profile of FAs in MetS is likely related to their dietary intake, MetS induced changes in FA endogenous metabolic transformation and genetic background. The direct measurement of desaturase and elongase activities is not easy to perform. Their estimated values representing the product-to-precursor ratio of corresponding FA content have been reported in several studies in the last decade, exploring the relation of FA composition with MetS [19,65,75,178,179,180,181,182]. Mayneris-Perxachs et al. (2014) described higher myristic (14:0), palmitic (16:0), POA, and DHGLA contents and indices of D9D and D6D activity in plasma lipids in subjects with cardiometabolic risk factors [178]. At the same time, the author reported the existence of a relationship between decreased plasma LA levels and estimated D5D activity with these factors. The adjustment for confounders did not diminish the significance of associations, except for DHGLA and estimated D9D and D5D activities. Muzsik et al. (2020) demonstrated an association between FA intake, erythrocyte PUFA profiles, and polymorphisms of FADS genes and MetS [65]. Comparing PUFA profiles in the erythrocyte membrane of postmenopausal women with and without MetS, they found that EPA, DPA n-3, and total n-3 PUFAs were lower in women with MetS. In addition, the authors reported that the D6D desaturase index was increased, whereas there was no significant difference in the estimated D5D activity. Although ALA content was not significantly different, the EPA/ALA and DPA n-3/ALA ratios were decreased in subjects with MetS. Moreover, the ratios of total n-3PUFAs/n-6 PUFAs as well as those of individual n-3 PUFAs with n-6 PUFA AA, EPA/AA, DHA/AA, and (DHA+EPA)/AA were also decreased in the MetS group. The polymorphisms of both *FADS1* and *FADS2* genes influenced EPA levels in erythrocyte membrane lipids, indicating that minor alleles are related to unfavorable FA profiles [65]. Recently, Sarac et al. (2023) observed that indices of D9D and D6D in erythrocytes were directly associated, while D5D activities were inversely associated with several indicators of cardiometabolic health (including fasting glucose levels, unfavorable TG and HDL-C profiles, and levels of global and visceral adiposity) in non-diabetic women [75]. At the same time, considering the erythrocyte PUFA pattern in this study, percentages of POA and DHGLA showed positive, whereas stearic acid (18:0) showed negative associations with the mentioned metabolic risk factors. However, after controlling for confounders, the level of adiposity was the most significant predictor of the desaturase activities and FA levels and mediated their association with other components of MetS. The levels of ALA, EPA, and DHA were not associated with the MetS components, while levels of AA were positively associated with some indices of adiposity [75]. The meta-analysis by Fekete et al. (2015) that examined obesity-induced changes in FA profiles demonstrated that estimated activities of D9D and D6D were increased, whereas the D5D index decreased, leading to increased DHGLA levels in different lipid pools in obese subjects [179]. Abdominal obesity is a key factor of MetS, and it is expected that changes in endogenous PUFA metabolism in MetS are like those in obesity. However, the metabolic changes in obesity and MetS are not the same. The influence of dietary FA intake and status on the induction of metabolic changes in obesity has been explored, contrasting the groups of obese subjects with and without associated metabolic changes. Bermúdez-Cardona et al. (2016) evaluated FA profiles in plasma phospholipids, comparing three groups of young people that were of appropriate weight, obese without MetS, and obese with MetS [180]. The lowest portion of LA was found in the MetS group. Moreover, DHGLA was significantly increased in the MetS group compared to the appropriate-weight group, whereas there were no significant changes when comparing the group of obese with the appropriate-weight subjects. Considering desaturases, the D9D and D6D indices were the highest, whereas the D5D index was lowest in the MetS group. Moreover, some metabolic changes can be found in both obese and non-obese subjects. In a recent study, dietary intake and status of PUFAs in plasma and erythrocytes were explored in four groups of subjects: metabolically unhealthy non-obese (MUHNO) and obese (MUHO) subjects, and metabolically healthy non-obese (MHNO) and obese (MHO) subjects [181]. Metabolically unhealthy subjects (both MUHNO and MUHO) had a higher energy intake of n-6 PUFAs, lower intake of n-3 PUFAs, and a higher ratio of n-6/n-3 PUFAs in their dietary intake, compared to MHNO and MHO subjects [181]. The key finding considering endogenous PUFA metabolism was that the MHO group had the higher mean EPA/AA ratio and estimated D6D and elongase 5 activity in plasma phospholipids, compared to MHNO. Low intake of n-3 PUFA is directly associated with metabolic risk factors and obesity, but endogenous metabolic transformation could play an important role in the development of metabolic changes in obese subjects. The differences between metabolically unhealthy (MU) and healthy (MH) phenotypes have also been explored by Svedeson et al. (2020), classifying subjects with ≥4 out of 5 criteria (increased concentrations of TG, TC, LDL-C, and HbA1c, and low HDL-C levels) to the MU group [182]. The most important finding of this study was that MH subjects had lower estimated D9D indices in the whole blood, while the value of the D6D activity index was higher compared to the MU group, which contrasts with previous findings. In summary, the estimated desaturase activities could be further tested as novel biomarkers of metabolic health. Moreover, elongase indices are not well-studied, and up to now, it has been revealed that they (elongases 6, 5, and 2) could also be altered in patients with impaired fasting glycemia [183].

## 5. Short Overview of the Mechanisms of n-3 PUFA Action

As structural components of membranes, n-3 PUFAs are primarily incorporated in phospholipids, but also in sphingolipids and plasmalogens, as well as in TG in adipose cells. The PUFA composition of plasma lipoproteins and membranes of erythrocytes, platelets, leucocytes, and compact tissues can be modified in a relatively short time (from days to weeks). The FA profiles in membranes affect the function of organelles, cells, and organs. The total percentage of n-3 PUFAs is up to 10% of total fatty acids in cell membranes. DHA is the most abundant n-3 PUFA in cell membranes, present in all tissues and accumulating in the retina, sperm, cerebral cortex, erythrocytes, spleen, liver, muscles, and heart, while its abundance in adipose tissue is the lowest [139]. ALA is present in minute amounts in adipose tissue, epithelium, heart, muscle, and erythrocytes. Still, it is the predominant n-3 FA in adipose tissue, accounting for 1% of the FAs stored as TGs. An overview of the mechanisms of n-3 PUFAs is presented in Figure 3. 

The incorporation of n-3 PUFAs into membranes of cells and organelles can influence their biophysical properties, including fluidity, thickness, and deformability. Moreover, n-3 PUFAs have been shown to change the properties (size and composition) of lipid rafts that are functional microdomains of the membranes rich in cholesterol, sphingolipids, and acyl chains of saturated fatty acids. These changes lead to modulation of protein–protein interaction, ion channel kinetics, signaling processes, and protein trafficking. The physiological effect of n-3 PUFAs is also exerted by direct interaction with membrane channels and some other proteins involved in intracellular signaling processes like protein G120, nuclear receptors, and transcription factors, thus influencing gene regulation. In recent years, the further mechanism of n-3 PUFAs action that involves the formation of their lipid mediators has attracted intense research interest. The relative proportions of DHA, EPA, ALA, and AA at the sn-2 position on membrane phospholipids determines their availability as precursors for the synthesis of lipid mediators after phospholipase A2 cleavage. The conversion of dietary ALA to long-chain n-3 PUFAs, bio-competition with LA, and biosynthesis of the ALA-, EPA- and DHA-derived lipid mediators, as well as the effect on the ECs system, could be of critical importance to the potentially favorable effects of ALA in MetS. 

## 6. Conclusions

In this narrative review, we briefly summarized current knowledge about the effects of ALA on the components of MetS and the ALA-induced changes in FA profiles, focusing on the transformation of ALA into its long-chain products in MetS. ALA seems to show favorable effects on adiposity, blood pressure, TG, and glucoregulation, as well as hepatic lipid accumulation and inflammation parameters in MetS subjects. The effects of the plant-derived ALA and marine-derived EPA and DHA may be synergic, and Mediterranean-type diets that are rich in both plant- and marine-derived n-3 PUFAs have shown positive effects on MetS. Recent studies have also demonstrated some unique effects of ALA on monocyte bioenergetics and glucoregulation. According to available data in the literature, dietary exposure to ALA leads to an increase in EPA and possibly DPA n-3 content, whereas long-chain n-6 PUFA levels could be decreased due to competition with LA for the same desaturation/elongation enzymes. This can lead to lower production of the AA-derived proinflammatory, obesogenic, and pro-insulin resistance metabolites of AA, including oxylipins and ECs. The efficiency of the ALA conversion may be enhanced by dietary changes, and recent studies mainly focus on the effects of polyphenols and zinc. However, the high dietary LA content, typical for the modern Western-type dietary pattern, can reduce the conversion rate of ALA. An important finding is that a hypoenergetic diet may lead to immediate oxidation of ALA, and, therefore, lower accumulation in cell membranes and conversion to long-chain n-3 PUFAs. Metabolic syndrome is found to induce changes in the bioconversion of ALA, with the increased estimated activities of D9D and D6D and possibly decreased activity of D5D. These alterations of endogenous metabolic transformation of ALA could be reflected by a high AA/EPA ratio in MetS. The changes in endogenous metabolism of PUFAs are similar but not completely the same as those in simple obesity, with the desaturases’ indices being separately associated with both obesity and metabolic disorders. The oxylipins derived from n-3 PUFAs can have anti-inflammatory effects, even those directly derived from ALA; however, the other effects of the ALA-derived C18 oxylipins are not well characterized, and more studies are needed. At present, it is hard to distinguish the direct effects of ALA from the effects of its metabolites. At the same time, analyzing a new source of ALA, camelina oil, it was found that it can reduce insulin sensitivity probably due to the high content of gonadic acid in MetS; therefore, the effects of ALA supplementation can be strongly dependent on the plant source of ALA, which needs to be considered when drawing conclusions or designing studies. In summary, the decrease in the n-6/n-3 ratio in the diet may be the primary measure in MetS to ensure a higher conversion of ALA to EPA, and possibly DHA. Further studies are needed to test whether some combination of ALA with other dietary components (currently, the primary candidates are zinc and polyphenols due to their effects on desaturases) could increase the conversion rate. Moreover, studying the profiles of different oxylipins in the circulation and different tissues (liver, adipose tissue, muscle) and their biological effects after dietary interventions with ALA in obesity and MetS may lead to the development of a personalized approach concerning ALA needs in MetS. 

## Figures and Tables

**Figure 2 ijms-25-04909-f002:**
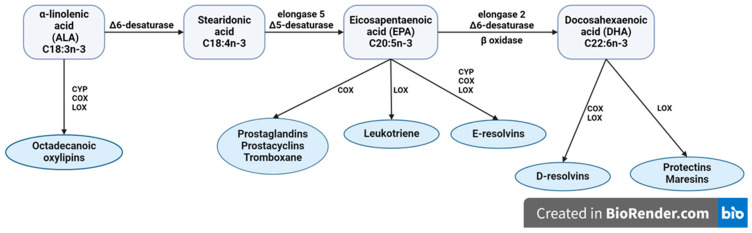
The metabolic conversation of α-linolenic acid (ALA) to long-chain polyunsaturated fatty acids of n-3 family through series of elongation and desaturation steps in the metabolic pathway; ALA and its metabolites eicosapentaenoic (EPA) and docosahexsaenoic (DHA) are precursors for synthesis of several classes of oxylipins that show anti-inflammatory and/or pro-resolving effects. CYP, cytochrome P450 oxidase; COX, cyclooxidase; LOX, lipooxidase.

**Figure 3 ijms-25-04909-f003:**
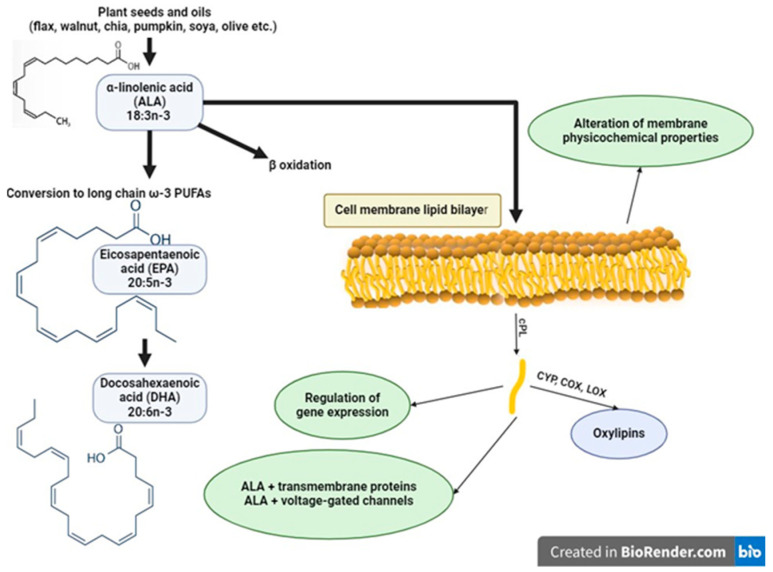
An overview of cellular mechanisms of α-linolenic acid action. Dietary intake of the acid (some seeds, nuts and vegetable oils are rich sources of ALA) can be (1) β-oxidized and thus used for production of metabolic energy, (2) converted to long-chain n-3 PUFAs, and (3) incorporated into the phospholipid bilayer of cells and membrane of organelles. The ALA presence in membranes determines its physicochemical properties (fluidity and biophysics, size, and composition of lipid rafts and caveolae) and modifies the functionality of different ion channels and other proteins in the membrane. ALA and its oxygenated products regulate gene expression through binding to nuclear receptors and transcription factors. Release from phospholipids by cytosolic phospholipase is the first step that leads to synthesis of oxylipins by oxygenation enzymes (COX—cyclooxygenase, LOX—lipooxygenase, CYP—cytochrome P450).

**Table 1 ijms-25-04909-t001:** List of recent studies exploring the association between ALA and MetS.

Authors	Type of Study	Participants	Follow-Up Period (years)	Duration of Clinical Trial (wk)	Source of ALA in Treatment Group	ALA Dose	Observed Outcomes
Flannagan et al. [59]	Cross-sectional study	468 parents, 201 children aged 7–12	2 y				In Adults, MetS prevalence is inversely associated with adipose tissue ALA and GLA.
Ding et al. [60]	Prospective study	2754 participants, men and women	8.8 y				↓ ALA and GLA, ↓ risks of MetS components (HTG, hypertension, and low HDL cholesterol)
Ma et al. [61]	Prospective study	1245 men and women	6 y				↑ ALA and GLA showed positive associations with a 6-year risk of developing incident ↑ MetS
Julibert et al. [62]	Cross-sectional study	6560 men and women	6 y				Participants in the highest quintile of total dietary fat intake showed higher intake of PUFA, MUFA, SFA, TFA, LA, ALA and ω-3 FA.
Wu et al. [21]	Randomized, controlled trial	283 men and women	12-week intervention	12-week intervention	Lifestyle counseling (LC), LC + flaxseed (LCF), LC + walnuts (LCW)	LCF 30 g/d LCW 30 g/d	16.7% reduction in MetS was observed. A low-intensity LC program could be useful in MetS. Flaxseed and walnut supplementation may ameliorate/improve central obesity.
Baxheinrich et al. [22]	Randomized, controlled trial	81 men and women	6 m	6 m	Rapeseed oil-rich diet (RO group) control diet with olive oil (OO group)	RO 3.5 g/dOO 0.78 g/d	↑ intakes of MUFA and ALA may be a practical approach for long-term dietary treatment in patients with metabolic syndrome, leading to weight reduction and an improvement in the overall cardiovascular risk profile.
Al Abdrabalnabi et al. [63]	Randomized clinical trial	625	2 y	6 m	Walnut group 30, 45, or 60 g of walnuts per day.	1, 1.5, or 2 oz. or ~15% of energy	No effect on MetS status or any of its components, although the walnut group tended to have lower blood pressure.
Ngo Njembe et al. [19]	Randomized, controlled trial	24 men and women	3 m	3 m	2 eggs enriched with ALA, DHA, RmA, and PunA	105.19 ± 4.04 mg/egg	A significant reduction in abdominal obesity, without improving other components of MetS, including glycaemia-associated parameters.
Egert et al. [20]	Randomized, controlled trial	81	26-wk intervention	26-wk intervention	Hypoenergetic diet high in ALA	3.4 g/d	Daily intake of 3.4 g of ALA did not increase serum phospholipid ALA or EPA. Additionally, dietary ALA was unable to compensate for a decrease in serum phospholipid DHA.
Bellien et al. [64]	Controlled randomized study	81	6 m	6 m	Camellia oil	1.5 g/d	ALA did not improve vascular function but adversely affected glucose metabolism in hypertensive patients with metabolic syndrome.
Muzsik et al. [65]	Case-control study	131 women	2 y				MetS is associated with lower levels of FAs that have a protective effect on cardiometabolic health.

ALA: α-linolenic acid; MetS: metabolic syndrome; GLA: γ-linolenic acid; HTG: hypertriglyceridemia; PUFA: polyunsaturated fatty acids; MUFA: monounsaturated fatty acids; SFA: saturated fatty acids; TFA: trans-fatty acid; LA: linolenic acid; FA: fatty acid; DHA: docosahexaenoic acid; RmA: rumenic acid; PunA: punicic acid; EPA: eicosapentaenoic acid; **↑** stands for increased.

**Table 2 ijms-25-04909-t002:** Efficiency of ALA conversion to its long-chain PUFA products.

Authors	Participants	Sex Differences Observed	Duration (wk)	Source of ALA	ALA Dose	Observed Outcomes
Burdge and Wootton [66]	6 women		21 d	[U^−13^C] ALA	700 mg	EPA 21%, DPA 6% and DHA 9%
Burdge et al. [67]	6 men		21 d	[U^−13^C] ALA	700 mg	ALNA to DHA was either very low or absent.
Greupner et al. [68]	19 men		12 wk	22.3 g of linseed oil	14.0 ± 0.45 g day^−1^	The intake of ALA is not a sufficient source for the increase in EPA + DHA in subjects on a Western diet.
Petrovic-Oggiano et al. [69]	18 men and women		4 wk	Walnuts	56 g	Plasma phospholipids ALA, EPA (10% ↑) and total n-3PUFA were increased.
Kuhnt et al. [70]	154 men and women		8 wk	Echium oil (EO) and Linseed oil (LO)	EO (5 g ALA + 2 g SDA) LO (5 g ALA)	Daily intake of STE-containing EO is a better supplement than LO for increasing EPA and DPA in blood. Neither EO nor LO maintained blood DHA status in the absence of fish/seafood consumption.
Barceló-Coblijn et al. [71]	62 men		12 wk	Flax oil capsules	1.2, 2.4, or 3.6 g flax oil/d	ALA-enriched supplements for 12 wk was sufficient to elevate erythrocyte EPA and DPA content.
Burdge et al. [72]	14 men		2 m	Spread	10 g/d	Raised plasma triacylglycerol-EPA and -DPA concentrations and phosphatidylcholine-EPA concentration.
Pawlosky et al. [73]	8 men and women		21 d	Isotope tracer of α-linolenate (d5-18:3n-3 ethyl ester)	1-g oral dose	About 0.2% of the plasma ALA was destined for synthesis of EPA, 63% of the plasma EPA was accessible for production of DPA, and 37% of DPA was available for synthesis of DHA.
Drobner et al. [74]	105 men and women	Higher in men than in women	12 wk	Different diets, each enriched with linseed oil	13–16 g	Daily intake of approx. 25 g linseed oil (=approx. 15 g ALA) leads to a significant increase in EPA concentrations and a simultaneous decrease in DHA concentrations in erythrocyte lipids.

ALA: α-linolenic acid; LA: linoleic acid; DPA: docosapentaenoic acid; DHA: docosahexaenoic acid; STE: stearidonic acid; EPA: eicosapentaenoic acid; CVD: cardiovascular diseases.

## Data Availability

Not applicable.

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
