# Peer review of "Current Insights into the Effects of Dietary α-Linolenic Acid Focusing on Alterations of Polyunsaturated Fatty Acid Profiles in Metabolic Syndrome"

_ijms, 2024, doi:10.3390/ijms25094909_

Round 1
Reviewer 1 Report
Comments and Suggestions for Authors
The manuscript titled "Current Insights into the Effects of Dietary α-linolenic Acid on Metabolic Syndrome: A Comprehensive Review of Polyunsaturated Fatty Acid Profile Alterations" aims to consolidate existing literature regarding the effects of α-linolenic Acid (ALA) on metabolic syndrome (MetS). The authors concentrate on elucidating the impact of dietary factors and endogenous metabolism of ALA on the alteration of PUFA profiles, which in turn may affect the synthesis of oxylipins, among other potential mechanisms. The manuscript, however, requires substantial revisions. The level of novelty is limited, and the authors neglect to engage deeply with the referenced papers and subjects. The overall narrative remains largely descriptive. Below are specific points for improvement
Specific comments:
1. The abstract requires significant refinement, particularly regarding clarity of rationale and articulation of the manuscript's novelty.
2. Improving the introduction requires addressing its vagueness and identifying a clear gap that the paper aims to fill. Moreover, providing more detailed information and mechanisms will enhance the introduction.
3. Improving Figure 1 requires enhancing its quality and depth to effectively convey the metabolic syndrome diagnostic criteria and the effects of α-linolenic acid (ALA), along with pertinent mechanisms of action.
4. To enhance Figure 2 and underscore the novelty and rationale of the review, integrating mechanisms of action, novel pathways, or biological effects related to α-linolenic acid (ALA) is crucial.
5. To improve Figure 3 and enhance clarity, it's essential to streamline the text and optimize visual elements for better comprehension.
4. Overall the paper is mostly descriptive without a in-depth analysis of effects nor discussion of mechanisms. That is clear on the discussion section in which there is a description and not an input from the authors.
Author Response
We are thankful to the reviewers for their constructive comments that helped us to further improve our manuscript. We considered all the comments and replied to them accordingly. All changes in the manuscript are marked as yellow text, except the English editing changes. Please note the point-by-point through responses below.
Reviewer 1
The manuscript titled "Current Insights into the Effects of Dietary α-linolenic Acid on Metabolic Syndrome: A Comprehensive Review of Polyunsaturated Fatty Acid Profile Alterations" aims to consolidate existing literature regarding the effects of α-linolenic Acid (ALA) on metabolic syndrome (MetS). The authors concentrate on elucidating the impact of dietary factors and endogenous metabolism of ALA on the alteration of PUFA profiles, which in turn may affect the synthesis of oxylipins, among other potential mechanisms. The manuscript, however, requires substantial revisions. The level of novelty is limited, and the authors neglect to engage deeply with the referenced papers and subjects. The overall narrative remains largely descriptive. Below are specific points for improvement.
Response: Thank you for the opportunity to revise our manuscript considering your suggestions. We hope that the quality of the current version of the manuscript is improved and that you will find it more suitable for publication.
Specific comments:
- The abstract requires significant refinement, particularly regarding clarity of rationale and articulation of the manuscript's novelty.
Response: Thank you for your suggestion. The abstract was carefully rewritten according to your suggestions, trying to emphasize better the manuscript novelty and is much better structured now.
- Improving the introduction requires addressing its vagueness and identifying a clear gap that the paper aims to fill. Moreover, providing more detailed information and mechanisms will enhance the introduction.
Response: Thank you for your valuable comment. We introduced changes in the introduction section of the manuscript, in line with the suggestions of all reviewers. We emphasized in the introduction that we have given an update on the published literature on that topic, and introduced some new mechanisms that were not previously covered in review articles (lines 114-125).
- Improving Figure 1 requires enhancing its quality and depth to effectively convey the metabolic syndrome diagnostic criteria and the effects of α-linolenic acid (ALA), along with pertinent mechanisms of action.
Response: Thank you for your useful suggestion. The new version of Figure 1 was prepared, and we hope that will better represent the potential beneficial effects of ALA on metabolic syndrome.
- To enhance Figure 2 and underscore the novelty and rationale of the review, integrating mechanisms of action, novel pathways, or biological effects related to α-linolenic acid (ALA) is crucial.
Response: Thank you for the great suggestions. The new version of the picture is prepared, and a more detailed description is added.
We corrected the manuscript according to your suggestions, and we went more deeply into the analysis of the available literature data, giving more potential new mechanisms that could explain those findings (e.g., the role of endocannabinoids, the oxylipins that directly derive from ALA, the effects on insulin sensitivity, adipocyte function, hepatocyte function, inflammation, etc. trying to explain the found effects on the MetS components).
In the manuscript, we focused more on the topic of the direct effects of ALA-derived oxylipins, not only those of the long-chain n-3 PUFA ALA-derivatives, EPA and DHA, whose effects are more known and already more reviewed in literature.
The new paragraph about ALA mechanism of action in obesity-induced inflammation is written considering your and the suggestions of the Reviewer 3.
- To improve Figure 3 and enhance clarity, it's essential to streamline the text and optimize visual elements for better comprehension.
Response: Thank you for the comment, we hope that the new version of the picture and figure description will meet the standards.
- Overall the paper is mostly descriptive without a in-depth analysis of effects nor discussion of mechanisms. That is clear on the discussion section in which there is a description and not an input from the authors.
Response: Thank you for the comment, we hope that the quality of the new version of the manuscript is improved as we revised all sections according to reviewers’ suggestions.
We corrected the manuscript, and we went more deeply into the analysis of the available literature data, giving more potential new mechanisms that could explain those findings (e.g., the role of endocannabinoids, the oxylipins that directly derive from ALA, the effects on insulin sensitivity, adipocyte function, hepatocyte function, inflammation, etc. trying to explain the found effects on the MetS components).
Reviewer 2 Report
Comments and Suggestions for Authors
The present review summarizes the effects of α-linolenic Acid on
metabolic syndrome components.
The topic is interesting.
Comments
Title: The title is comprehensive. However, it can be shortened.
Abstract: Multiple tenses are used in the Abstract. Please try to use one or two tenses.
Introduction:
Line 53 “However, favorable effects have been observed in some recent studies [12-15]”. A brief report of the favorable effects shown in these studies would be useful to the reader.
Literature search and study selection: The authors describe the methodology used, but there is no information about the results of the research e.g. the number of papers selected, etc.
Impact of dietary α-linolenic acid on metabolic syndrome:
Lines 102-104 “…conditions like chronic inflammation, hepatic steatosis, impaired kidney function, obstructive sleep apnea, polycystic ovary syndrome and hyperuricaemia…”. A lot more abnormalities (such as hypercoangulation, cancer, etc) are linked to MS and thus they could be added. Moreover, relevant citations are lacking.
In Line 173 the authors mention that “Regarding impact of dietary ALA on cardiometabolic factors, favorable effects on adiposity, TG levels and hypertension have been observed in some, but not confirmed in all studies”.
Moreover, in the concusion section the authors mention that “ALA seems to show favorable effects on adiposity, blood pressure and possibly triglyceride levels considering cardiometabolic risk factors as well as inflammation parameters in MetS subjects”
However, in Figure 1, it is mentioned that there is an increase in TG level and a decrease in HDL level, effect that is definitely not favorable. In any case, there is no citation for Figure 1 and thus it is not clear which is the source of the green bars (effects of ALA). In general, it seems that Figure 1 is misleading and it has to be corrected (in order to be in line with the main text) or removed.
Line 221: Which glycemic markers? Markers having to do with glucose levels or markers having to do with insulin resistance? This is important; since the authors in Figure 1 summarize that, there is no significant effect. In any case, pre-diabetes and diabetes are not synonymous with insulin resistance. One can have insulin resistance and simultaneously be normo-glycemic, in cases that there is compensatory insulin secretion from the pancreas.
Conclusion:
In the conclusion section there are studies mentioned without relevant citation.
Line 622 The authors mention “…A new important finding…” Do they mean a recent finding in the published literature or a new conclusion derived from their review?
Line 634 “..The future results of this study may significantly contribute to better…” It is not clear what the authors mean by “future results of this study”.
Comments on the Quality of English Language
There are several syntactical /grammatical errors. For example:
The word beside is used instead of besides in several parts of the manuscript.
Line 30: “…is an essential fatty acids”. Please use acid instead of acids. Line 120 ”However, but no significant …” Please delete ” but”.
…etc
The whole manuscript has to be checked by a native English speaker.
Author Response
We are thankful to the reviewers for their constructive comments that helped us to further improve our manuscript. We considered all the comments and replied to them accordingly. All changes in the manuscript are marked as yellow text, except the English editing changes. Please note the point-by-point through responses below.
Reviewer 2
The present review summarizes the effects of α-linolenic acid on metabolic syndrome components. The topic is interesting.
Response. We thank the reviewer for his/her overall positive comment. In addition, we are grateful for the opportunity to refine our work in light of your suggestions. We are confident that incorporating the suggested changes will significantly enhance the quality and clarity of the manuscript.
Comment 1. Title: The title is comprehensive. However, it can be shortened.
Response 1. Thank you for the suggestion. The title is now modified and shortened.
Comment 2. Abstract: Multiple tenses are used in the Abstract. Please try to use one or two tenses.
Response 2. Thank you for the comment. We carefully checked and the abstract is now completely rewritten and better structured.
Comment 3. Line 53 “However, favorable effects have been observed in some recent studies [12-15]”. A brief report of the favorable effects shown in these studies would be useful to the reader.
Response 3. Thank you for a valuable observation. A brief description summarizing favorable effects is added.
Comment 4. Literature search and study selection: The authors describe the methodology used, but there is no information about the results of the research, e.g. the number of papers selected, etc.
Response 4. Thank you for the suggestion. Accordingly, we added more detailed description about results of literature search and included the number of selected papers.
Comment 5. Lines 102-104 “…conditions like chronic inflammation, hepatic steatosis, impaired kidney function, obstructive sleep apnea, polycystic ovary syndrome and hyperuricaemia…”. A lot more abnormalities (such as hypercoangulation, cancer, etc) are linked to MS and thus they could be added. Moreover, relevant citations are lacking.
Response 5. Thank You for the suggestion. We expanded the paragraph on the association of metabolic syndrome with the risk factors and diseases. We also added the relevant citations and more definitions of diagnostic criteria for MetS.
Comment 6. In Line 173 the authors mention that “Regarding impact of dietary ALA on cardiometabolic factors, favorable effects on adiposity, TG levels and hypertension have been observed in some, but not confirmed in all studies”.
Moreover, in the concusion section the authors mention that “ALA seems to show favorable effects on adiposity, blood pressure and possibly triglyceride levels considering cardiometabolic risk factors as well as inflammation parameters in MetS subjects”
However, in Figure 1, it is mentioned that there is an increase in TG level and a decrease in HDL level, an effect that is definitely not favorable. In any case, there is no citation for Figure 1 and thus it is not clear which is the source of the green bars (effects of ALA). In general, it seems that Figure 1 is misleading and it has to be corrected (in order to be in line with the main text) or removed.
Response 6. Thank you for the comment regarding Figure 1. We apologize for the mistake that was made considering α-linolenic effects on HDL-levels. The new Figure 1 is prepared and relevant references are cited in Figure legend.
Comment 7. Line 221: Which glycemic markers? Markers having to do with glucose levels or markers having to do with insulin resistance? This is important; since the authors in Figure 1 summarize that, there is no significant effect. In any case, pre-diabetes and diabetes are not synonymous with insulin resistance. One can have insulin resistance and simultaneously be normo-glycemic, in cases that there is compensatory insulin secretion from the pancreas.
Response 7. Thank you for a valuable observation. We strongly agree that a more detail description about pre-diabetes and diabetes as a component of MetS is needed. The section about glucoregulation is accordingly modified in line with your suggestions.
Comment 8. Line 622 The authors mention “…A new important finding…” Do they mean a recent finding in the published literature, or a new conclusion derived from their review?
Response 8. Thank you for the comment, we meant a recent finding in the published literature and corrected the text in conclusion paragraph.
Comment 9. Line 634 “.The future results of this study may significantly contribute to better…” It is not clear what the authors mean by “future results of this study”.
Response 9. Thank you for the comment. We modified the sentence to make it clearer.
Comment 10. There are several syntactical /grammatical errors. For example: The word beside is used instead of besides in several parts of the manuscript. Line 30: “…is an essential fatty acids”. Please use acid instead of acids. Line 120 ”However, but no significant …” Please delete ” but”.…etc The whole manuscript has to be checked by a native English speaker.
Response 10. The whole manuscript is carefully checked by the authors more proficient in English, as well as Proofreading and Grammarly programs.
Reviewer 3 Report
Comments and Suggestions for Authors
Journal : IJMS (ISSN 1422-0067)
Manuscript ID ijms-2902226
This review addresses a crucial topic within human physiology, specifically metabolic syndromes. The focus revolves around the intriguing exploration of the impact of α-linolenic acid ingestion on metabolic syndromes, encompassing parameters such as EPA, DHA, AA/EPA ratio, and delta-6-desaturase activity.
1. The abstract needs to be revised with a structured sequence of ideas: starting with the ingestion of ALA, followed by its intestinal absorption and stability, and subsequently examining its impact on specific metabolic markers and inflammatory indices.
2. The introduction is well-organized and effectively written.
3. Lines 97-99 ; Elevated blood pressure or a glucose level exceeding ≥100 mg/dl should not be solely relied upon as indicators of metabolic syndromes; careful assessment is necessary. Consider incorporating clinical indices such as DHD/LDL ratios for a more comprehensive evaluation.
4. Lines 102-103 ; Certainly! When referencing information, it's essential to include the appropriate citations. If you have specific information or statements you'd like me to provide references for, please provide the details, and I'll do my best to assist you in finding suitable references.
5. The presentation of Figure 1 is suboptimal, as it contains disorganized information without proper references. Consider revising and improving the figure by providing clear structure and citing relevant references for the information presented.
6. Line 111 « Beside weight loss, the American diabetic association (ADA) has recommended a diet with low…Meaningless sentence, to revise
7. Based on line 178, it suggests that ALA is susceptible to oxidation, potentially leading to toxic effects. Therefore, it is crucial to dedicate a paragraph to discuss the stability of this lipid, examining factors that may influence its susceptibility to oxidation and the potential consequences of such instability.
9. Referring to lines 196-198, it's essential to determine whether the ingestion of ALA results in a reduction in HDL-C levels. Consider revising to clarify whether this effect is a reduction or a regulation of HDL-C levels.
10. Paragraph 3.3.4, which examines the impact of LA on glucoregulation in four lines, falls short in highlighting the significance of hyperglycemia and ALA D in metabolic diseases. It requires revision to emphasize the crucial role of these factors in metabolic disorders.
11. Paragraph 3.4.1 necessitates a thorough review of the mechanisms through which ALA acts against inflammation. Consider revising to provide a comprehensive examination of how ALA functions in mitigating inflammatory processes.
12. Include a paragraph that delves into the toxic effects of this functional food, elaborating on potential adverse impacts and considerations associated with its consumption.
13. Integrate an abstract figure illustrating the diverse mechanisms through which ALA interacts with metabolic diseases, encompassing aspects related to obesity, diabetes, hypertension, and other relevant factors.
Comments on the Quality of English LanguageCertainly! If you could provide specific sentences or paragraphs that you find incomplete or lacking scientific clarity, I'd be happy to help you revise them for better coherence and accuracy.
Author Response
We are thankful to the reviewers for their constructive comments that helped us to further improve our manuscript. We considered all the comments and replied to them accordingly. All changes in the manuscript are marked as yellow text, except the English editing changes. Please note the point-by-point through responses below.
Reviewer 3
This review addresses a crucial topic within human physiology, specifically metabolic syndromes. The focus revolves around the intriguing exploration of the impact of α-linolenic acid ingestion on metabolic syndromes, encompassing parameters such as EPA, DHA, AA/EPA ratio, and delta-6-desaturase activity.
Thank you for an overall positive comment. We are grateful for the opportunity to revise our manuscript considering your suggestion.
- The abstract needs to be revised with a structured sequence of ideas: starting with the ingestion of ALA, followed by its intestinal absorption and stability, and subsequently examining its impact on specific metabolic markers and inflammatory indices.
Response: Thank you for your suggestion. The abstract was carefully rewritten according to your suggestions, trying to emphasize better the manuscript novelty and is much better structured now.
- The introduction is well-organized and effectively written.
Response: Thank you for your positive comment about the introduction section.
- Lines 97-99 ; Elevated blood pressure or a glucose level exceeding ≥100 mg/dl should not be solely relied upon as indicators of metabolic syndromes; careful assessment is necessary. Consider incorporating clinical indices such as DHD/LDL ratios for a more comprehensive evaluation.
Response: The reference of Grundy at al. (2005) represents the latest official diagnostic criteria used for diagnosis of Mets by the American Heart Association (AHA)/the National Heart, Lung, and Blood Institute (NHLBI) in the Europid population. Of course, there are some other definitions and proposed indices, including lipid ratios, but they are still not official. Therefore, in line with your suggestion, we improved the paragraph, by adding the different definitions of MetS, giving more on the lipid ratios as the indicators of MetS, and other features that are associated with MetS (marked with yellow in the text).
- Lines 102-103; Certainly! When referencing information, it's essential to include the appropriate citations. If you have specific information or statements you'd like me to provide references for, please provide the details, and I'll do my best to assist you in finding suitable references.
Response: Thank you for the suggestion, we added appropriate references to the manuscript text.
- The presentation of Figure 1 is suboptimal, as it contains disorganized information without proper references. Consider revising and improving the figure by providing clear structure and citing relevant references for the information presented.
Response: We prepared a new version of Figure 1 in line with your suggestions and added relevant references
- Line 111 « Beside weight loss, the American diabetic association (ADA) has recommended a diet with low…Meaningless sentence, to revise
Response: Thank you for a comment, the sentence is rewritten.
- Based on line 178, it suggests that ALA is susceptible to oxidation, potentially leading to toxic effects. Therefore, it is crucial to dedicate a paragraph to discuss the stability of this lipid, examining factors that may influence its susceptibility to oxidation and the potential consequences of such instability.
Response: Thank you for your suggestion. The paragraph considering oxidative stability and potential consequences of ALA instability is added in the text (lines from 41 to 61)
- Referring to lines 196-198, it's essential to determine whether the ingestion of ALA results in a reduction in HDL-C levels. Consider revising to clarify whether this effect is a reduction or a regulation of HDL-C levels.
Response: According to literature data, the ingestion of ALA does not reduce HDL-C cholesterol concentration. We clarified this in the text. Thank you for the comment.
- Paragraph 3.3.4, which examines the impact of LA on glucoregulation in four lines, falls short in highlighting the significance of hyperglycemia and ALA D in metabolic diseases. It requires revision to emphasize the crucial role of these factors in metabolic disorders.
Response: Thank you for your valuable observation. We added a more detail description about glucoregulation in metabolic diseases.
- Paragraph 3.4.1 necessitates a thorough review of the mechanisms through which ALA acts against inflammation. Consider revising to provide a comprehensive examination of how ALA functions in mitigating inflammatory processes.
Response: Paragraph 3.4.1 was rewritten according to your suggestions. We corrected also paragraph 4.4., focused on the effects of ALA derived oxylipins and their effects on inflammation.
- Include a paragraph that delves into the toxic effects of this functional food, elaborating on potential adverse impacts and considerations associated with its consumption.
Response: The paragraph considering oxidative stability of ALA that could produce adverse effects on health is added in the text. We also mentioned the adverse effects of the camelina oil.
- Integrate an abstract figure illustrating the diverse mechanisms through which ALA interacts with metabolic diseases, encompassing aspects related to obesity, diabetes, hypertension, and other relevant factors.
Response: We prepared the graphical abstract according to your suggestions.
Comments on the Quality of English Language
Certainly! If you could provide specific sentences or paragraphs that you find incomplete or lacking scientific clarity, I'd be happy to help you revise them for better coherence and accuracy.
The English language editing was performed. Please, feel free to give your suggestions, if you find that there are still issues with the English language editing.
Round 2
Reviewer 1 Report
Comments and Suggestions for Authors
The revised version has improved the quality of the manuscript
Reviewer 2 Report
Comments and Suggestions for Authors
The revised form of the manuscript has been improved.
Reviewer 3 Report
Comments and Suggestions for Authors
In my opinion, this revised manuscript is revised and the authors responded to my suggestions and the requested modifications were made and can therefore be accepted for publication